# Pseudo-Static Analysis on the Shifting-Girder Process of the Novel Rail-Cable-Shifting-Girder Technique for the Long Span Suspension Bridge

**Quan Pan, Zhuangpeng Yi \*, Donghuang Yan \* and Hongsheng Xu**

School of Civil Engineering, Changsha University of Science and Technology, Changsha 410114, China; panquan605@126.com (Q.P.); hongsheng74@126.com (H.X.)
* Correspondence: yizhuangpeng@163.com (Z.Y.); yandonghuang@126.com (D.Y.); Tel.: +86-731-85256066 (Z.Y. & D.Y.)

**Abstract:** The rail-cable-shifting-girder (RCSG) technique is a new erecting method for the main girders of the long span suspension bridge in rural mountain areas with poor transportation and no navigable rivers for carrying large components. The pseudo-static analysis of the shifting-girder process for this girder erecting technique is performed. The global mechanical model of the double-layer cable system in the shifting-girder process is established, by analytically modeling the main-cable, rail cable, and slings according to cable's basic assumptions. Based on the flexible cable theory, the main-cable segments are simulated as segmental catenary elements, the slings are considered as straight cable elements, the rail-cable segment that the shifting-girder trolley is moving on is simulated as multiple straight cable elements and other rail-cable segments are considered as single straight cable elements. The solving program is developed to obtain the pseudo-static responses including the forces and deflections of the shifting-girder system undergoing girder loads. Meanwhile, a global indoor reduced-scale model of shifting-girder system is designed to validate the presented theoretical results, by taking the Aizhai suspension bridge as engineering background. The results from the presented theoretical method match well with the measured experimental results of the indoor model test. The forces and deflections of the main-cable, rail-cable, and slings for the 21 working cases of erecting girder segments exhibit some specific distribution regularities. The presented theoretical method is able to correctly and effectively solve the pseudo-static responses of the RCSG system undergoing girder loads for the long span suspension bridge adopting the construction method of the RCSG technique.

**Keywords:** suspension bridge; girder; RCSG; pseudo-static analysis; model test

## 1. Introduction

The suspension bridge is a very useful and powerful transportation tool to go across super long barriers due to its excellent mechanical properties, and it has attracted the interest of many researchers. For example, the finite element modeling [1], the vibratory characteristics [2,3], wind-resistant designing [4], mechanical performance [5], health monitoring [6], and structural reliability [7] have been the research topics. In the construction stage [8,9] of a suspension bridge, the attention of researchers is also placed on the risk assessment [10], modal analysis [11,12], passive aeroelastic control [13], galloping [14], fluttering [15], and wind tunnel test assessment [16]. For the construction technique of this kind of super-long bridge, it should be particularly pointed out that in the rural mountains, through valleys, the segments of the steel box girder cannot arrive at the mountainous area, due to the poor transportation and no navigable rivers available for carrying large

components. Therefore, the main girder of a long-span suspension bridge at a mountain area usually adopts steel trussed structure with piece transportation and assembly at construction scene. The cable hoisting method [17], the bridge deck crane method [18], and the rail-cable-shifting-girder (RCSG) technique [19] are available in the construction of the steel truss stiffening girder for a long span suspension bridge in mountainous areas. For these technologies, the requirements on the girder erecting manners, hinge setting, rigid connection conversion, erection period, and girder height are different.

The RCSG technique is a new erection method for the stiffening girder, which has gradually become a major method suitable for the girder erection of a long-span suspension bridge in mountainous areas, through the global model test [19] and actual implementation in the Aizhai suspension bridge [20–22]. As shown in Figure 1, the shifting-girder system mainly composed of main-cable, sling, hanging saddle and rail-cable is a double-layer flexible suspension system. The initial shape and stiffness can be calculated from the equilibrium equation according to the pretension state and boundary conditions of the rail-cable, through the so-called shape finding analysis process [23,24].

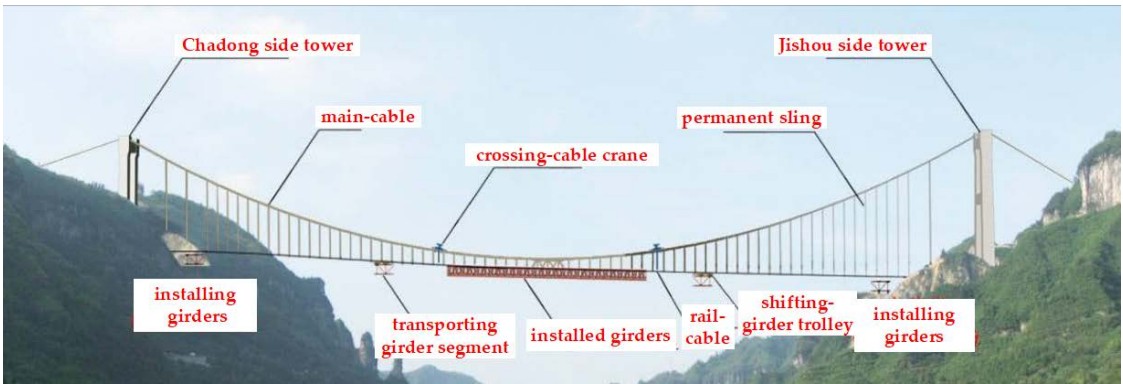

**Figure 1.** The shifting-girder system in the rail-cable-shifting-girder (RCSG) technique of the Aizhai suspension bridge during construction.

The main technical processes of the RCSG technique include: (1) the girder-carrying rail cable with horizontal pres-tension is set up at the bottom of the sling by taking the main-cable and permanent sling as supports; (2) the main-cable is connected to the sling by a saddle to assemble the steel truss girder segment at two sides, respectively; (3) a single girder segment will be placed below the permanent sling through the shifting-girder trolley in longitudinal transportation; (4) the steel truss girder segment will be lifted by the crossing-cable crane or other lifting equipment, shifting-girder trolley exits; (5) the girder segment is docked and the sling is pinned; and (6) each girder segments are constructed symmetrically on both sides from mid-point to both sides until the closure of the entire bridge.

Quite different from the traditional cable-carrying traffic system, the rail-cable in the RCSG system is hoisted at the lower end of the flexible sling. The small part of the dead weight of the shifting-girder trolley and girder segments is balanced by the horizontal pre-stressed forces of the rail-cable, while the main part of them transmit to the main-cable from the sling and then to the anchor system. Also, the supporting condition in the RCSG system is different from the general cable with rigid pylon supports, which relies on the fact that the span of the rail-cable is reduced due to the suspending and supporting of slings and a hanging saddle, and the RCSG system is a double-layer cable structure with multi-point flexible support. Moreover, because of the strong elastic resilience the double-layer cable system can ensure its lateral stability, thus the stiffness of the main-cable is far greater than that of the load-bearing cable of an ordinary rail transit and has a far greater load-carrying capability, e.g., the shifting-girder trolley of the Aizhai suspension bridge during construction can bear a 240 t load. Therefore, our research focused on the RCSG technique are new and many related mechanical problems need to be solved.

For the RCSG technique, the sling is the link between the main-cable and the rail-cable. The sling elongation is limited under the no-load condition, while the sling is undergoing alternating load in

the erection process of girder segments, and the vertical interaction force between the main-cable and rail-cable is discontinuous. Once the girder segment is installed, the sling force at this place reaches its maximum with a relatively stable value in the shifting-girder process. The sling is flexible and it extends under loads. This elongation must be considered in the refined analysis of the deformation coordination relation. Especially, the undergoing load of the sling is changing in the shifting-girder process, and its elongation cannot be ignored. The rail-cable is one part of the flexible system composed of the main-cable and sling, the large deformation must be considered, which brings the difficulty and complexity to the pseudo-static analysis and solution.

The shifting-girder trolley, as shown in Figure 2, is an important part of the shifting-girder system, which is composed of two pairs of tackle-block and moves on the rail-cable. The systematical responses are different when the shifting-girder trolley locates in different position with various inter-segmental lengths and dragging methods. In the process of the first pair tackle-block transiting from the flexible rail-cable to the rigid hanging saddle and then sliding from the hanging saddle to the rail-cable, the locally relative deformation of the rail-cable slowly increases. With the moving forward of the tackle-block by the dragging force, the relative deformation of the rail-cable is becoming bigger, and the tackle-block has the trend of declining quickly under the effect of its gravity. While the subsequent pair tackle-block start to climb the hanging saddle through the rail-cable, to slow down the movement of the tackle-block. In the design of longitudinal size of the shifting-girder trolley, the span of the two gravity centers of the shifting-girder trolley adopts half of the hanging sling spacing, to reduce the shock to the shifting-girder trolley. When the first pair tackle-block is climbing, the second one will move downhill. Conversely, when the first pair is downhill, the second pair will move uphill, ensuring the consistency of the drag force of the system and preventing a crash.

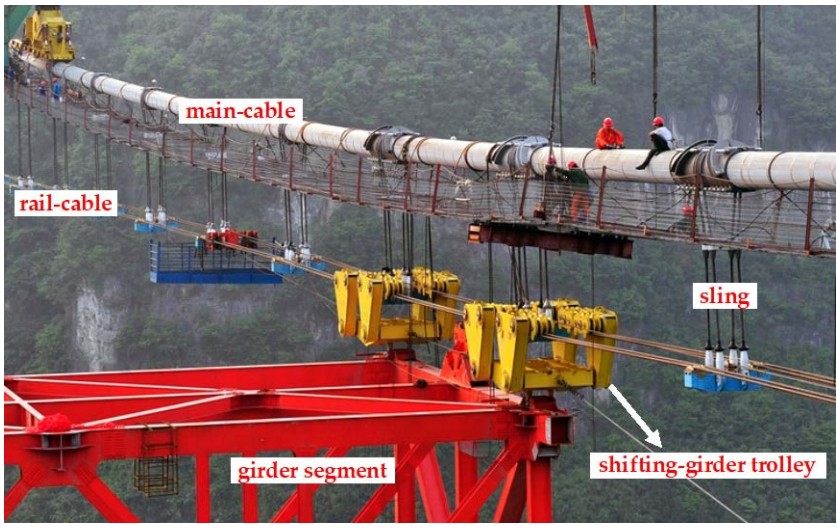

**Figure 2.** The shifting-girder trolley in the RCSG system.

The global transportation of the girder segment is realized in the process of the RCSG technique, which greatly reduces the air operating works and erecting time. For instance, the installation period of the 1000.5 m girders of the Aizhai suspension bridge took only 2.5 months, saving about 2000 t of steel used for the steel truss girder. With this new technology, the construction cost is lower, and the construction qualities of the girder are improved. Therefore, this technique provides a new, economical, and efficient erection method for the girder of the long span suspension bridge in rural mountainous areas with poor transportation and no navigable rivers.

The stiffness of the shifting-girder system increases because of the pre-tension of the rail-cable. When the force of the rail-cable is larger, the vertical stiffness is greater, the up/down angle of the shifting-girder trolley on the rail-cable is smaller, and the moving of the girders is smoother. This is much different from some research cases of the main-cable in a suspension bridge, where only the

main-cable's gravity and sling force are considered. If the traditional finite element method is adopted to investigate this shifting-girder system, the loads of shifting-girder trolley and girder can only add in the cable nodes when the rail-cable line is divided into segment units. Also, if the rail-cable segments are not divided into very-thin cable units, the relative deformation between the rail-cable segments cannot be calculated, i.e., the pre-tension force and diameter of the rail-cable cannot be determined. However, too many refined rail-cable elements will increase the computational costs, and there is a risk of non-convergence. Moreover, in the analysis of the traditional cable truss and the ordinary double-layer cable system, the connection rod between the main-cable and stable-cable is assumed to be rigid, the vertical displacements of two cables are the same, and the link rod is assumed to be continuous. This is different from the RCSG system and some important mechanical properties cannot be considered. Therefore, there is still room for optimization [25,26] and simplification of the RCSG technique to deal with the aforementioned insufficiencies. Particularly, the form-finding analysis [27] of the RCSG system undergoing no loads during the erection of the long span suspension bridge was performed in our previous paper.

In this paper, the attention is placed on the pseudo-static responses (i.e., forces and deflections) of the RCSG system undergoing girder loads during the erection of girder segments. The global model of the shifting-girder system is established, by analytically modeling the main-cable, sling and rail-cable according to the basic assumption of flexible cable mechanics [28–31]. The presented method deals with the research difficulties, which cannot be considered in the traditional finite element method of the RCSG technique. Also, it can simplify the pseudo-static analysis of the RCSG system. These constitute the innovation and motivation of our work. The model is proposed after the accomplishment of shape finding of no-loading cable [27]. The related loads are simulated and the solving program is compiled. For the RCSG technique in the construction of the long span suspension bridge, an indoor model test is also proposed to validate the correctness of the presented theoretical method.

## 2. Mechanical Model of Shifting-Girder System Undergoing Girder Loads

### 2.1. Basic Assumptions

The shifting-girder process undergoing girder loads is defined as the shifting-girder trolley carrying girder segments from the lifting position and moving it to the installed position, after the completion of the erection of the main-cable, sling, and cable clip, and the installation and tension of the rail-cable, as well as the lifting equipment at its position. To simplify the computation, the following assumptions [19] are introduced:

(1)　The flexible cable can only be tensioned, and compression and bending are not considered.
(2)　The stress-strain of the flexible cable satisfies Hooke's law.
(3)　The section area before and after deformation is unchanged to compute the tensile stiffness of the main-cable, rail-cable, and sling.
(4)　The main-cable, rail-cable, and sling are uniform in span length with a constant cross-sectional area and elastic modulus.
(5)　The friction between the rail-cable and hanging saddle is ignored, due to the fact that the hanging saddle is smooth and their relative deformations are small. The sling has no inclination. The weight of the steel truss girder and the shifting-girder trolley are known.
(6)　The contribution of the girder to the systematical stiffness is not considered in the shifting-girder process, i.e., all girder segments are hinged.

### 2.2. Global Mechanical Model

The global mechanical model of the RCSG system undergoing girder loads is established as shown in Figure 3, where the subscripts $z$, $g$, and $d$ in the variables, which respectively represent the main-cable, rail-cable, and sling, and $o$-$XY$ are the global Cartesian coordinates. Here, the rail-cable is divided into $m$ segments at the connected points with $m-1$ slings; and the main-cable is divided

into *n* segments according to actual condition, whose ends includes the connected points with *m*−1 slings. $A_{zi}$ (*i* = 0, 1, ... , *n*) are the node locations of the main-cable segments; $H_{zi}$, $V_{zi}$ (*i* = 0, 1, ... , *n*) are the horizontal and vertical force at the left section of node $A_{zi}$ for the main-cable, and $H_{z0}$, $V_{z0}$, $H_{zn}$, and $V_{zn}$ are the horizontal and vertical pretension forces of the main-cable at the left and right supports; $L_{zi}$, $S_{zi}$ (*i* = 1, 2, ... , *n*) are, respectively, the horizontal length and arc length of *i*th segment of main-cable after the installation of slings, and $h_{zi}$ (*i* = 1, 2, ... , *n*) are the vertical height differences between the two ends of the *i*th segment of main-cable; $L_z$ and $h_z$ are respectively the total horizontal span and vertical height difference between two end points of the main-cable, i.e., the left and right towers of suspension bridge. $B_{zj}$ (*j* = 0, 1, ... , *m*) are the node locations of rail-cable segments; $H_{gj}$, $V_{gj}$ (*j* = 0, 1, ... , *m*) are the horizontal and vertical force at the left section of node $B_{zj}$ for the rail-cable, and $H_{g0}$, $V_{g0}$, $H_{gm}$, $V_{gm}$ are the horizontal and vertical pretension forces of the rail-cable at the left and right supports; $L_{gj}$, $S_{gj}$ (*j* = 1, 2, ... , *m*) are the horizontal length and arc length of *j*th segment of rail-cable after the installation of slings, and $h_{gj}$ (*j* = 1, 2, ... , *m*) are the vertical height differences between the two ends of the *j*th segment of rail-cable; $L_g$ and $h_g$ are, respectively, the total horizontal span and vertical height difference between the anchor points at two ends of the rail-cable. $S_{dj}$ (*j* = 1, 2, ... , *m*-1) are the vertical or arc lengths of the *j*th sling after the installation of all slings; $q_1$, $q_2$, and $q_3$ are, respectively, the self-weight per unit length of the main-cable, rail-cable, and sling. The weights of girder segment and shifting-girder trolley are equivalent to concentrated forces on the two nodes of the rail-cable. Due to the standard girders adopting open-type structure, the distributed loads on two nodes are unequal. The two node-loads are respectively expressed by $W_1$ and $W_2$, the load ratio on lifting points is 1.7:1 [19], i.e., $W_1 = 1.7 \times W_2$. The weights of girder and shifting-girder trolley are known, therefore, $W_1$ and $W_2$ are determined.

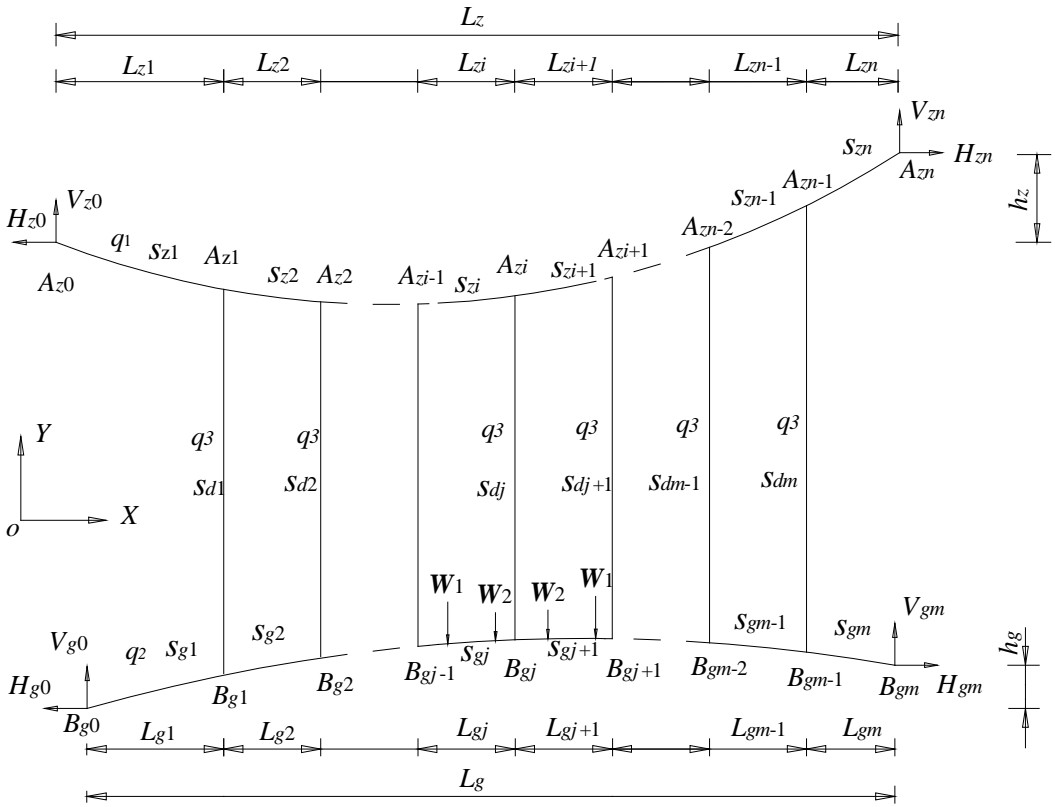

**Figure 3.** Mechanical model of the shifting-girder system undergoing girder loads.

To describe the shifting-girder process, an inter-node model is built shown in Figure 4. Under loading conditions, each segment of main-cable between two slings is further discretized into two main-cable segments $A_{zi}$-$A_{zi+1}$ and $A_{zi+1}$-$A_{zi+2}$. The cable equations [19] for these two

segments are established by using the segment catenary element. Then the lengths and heights $L_{zi}$, $h_{zi}$, $L_{zi+1}$ and $h_{zi+1}$ for these two segments are determined as follows:

$$L_{zi} = \frac{H_{zi}S_{zi}}{E_1A_1} - \frac{H_{zi}}{q_1}\{\ln(V_{zi} + \sqrt{H_{zi}^2 + V_{zi}^2}) - \ln[V_{zi} - q_1S_{zi} + \sqrt{H_{zi}^2 + (V_{zi} - q_1S_{zi})^2}]\}, \tag{1}$$

$$h_{zi} = \frac{q_1S_{zi}^2 - 2V_{zi}S_{zi}}{2E_1A_1} - \frac{1}{q_1}[\sqrt{H_{zi}^2 + V_{zi}^2} - \sqrt{H_{zi}^2 + (V_{zi} - q_1S_{zi})^2}], \tag{2}$$

$$L_{zi+1} = X_{zi+1} - X_{zi}, \tag{3}$$

$$h_{zi+1} = Y_{zi+1} - Y_{zi}, \tag{4}$$

where $E_1$, $A_1$ are elastic modulus and cross-sectional area of the main-cable, $X_{zi}$, $Y_{zi}$, $X_{zi+1}$, $Y_{zi+1}$ are the Cartesian coordinates for the two nodes of main-cable in the main-cable in *o-XY*. Next, according to equilibrium condition of node $A_{zi}$ and sling, the balance equations are established as follows:

$$H_{zi-1} = H_{zi}, \tag{5}$$

$$V_{zi} = V_{zi-1} - P_{zj} - q_1S_{zi}, \tag{6}$$

where $P_{zj}$ is the sling force of *j*th sling who connected with node $A_{zi}$ at the top.

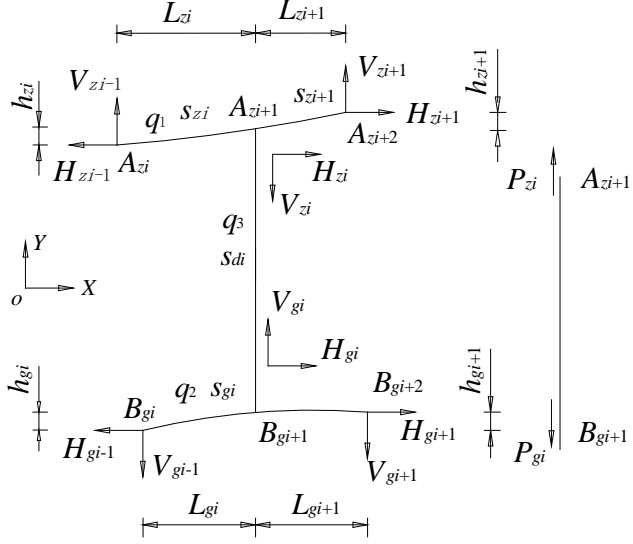

**Figure 4.** Inter-node model of the shifting-girder system undergoing girder loads.

Based on Assumption (5), by regardless of the slant of the sling, the coordination relationships between the force and deformation are written as:

$$P_{zj} = P_{gj} + q_3S_{dj} + W, \tag{7}$$

$$Y_{zi} - Y_{gj} = \frac{P_{gj}}{E_3A_3} + S_{dj}, \tag{8}$$

where $P_{gj}$ is the sling force increment of the *j*th sling; $W$ is the constant weight of the saddle; $S_{dj}$ is the unstressed cable length of the sling, $E_3$, $A_3$ are elastic modulus and cross-sectional area of the sling.

To analyze the global and local responses of the shifting-girder system, four straight cable elements [19] are adopted to simulate the area of the rail-cable segments that the shifting-girder trolley is moving on. To ensure working points of the wheel pairs of the shifting-girder trolley they are uniformly placed on the nodes of the proposed straight cables, the rail-cable segments at other positions

are simulated by one straight cable element. The discrete chart of the rail-cable segment is shown in Figure 5, where $B_{x1}$, $B_{x2}$, and $B_{x3}$ are the three added nodes to divide one rail-cable segment into four straight cable elements. Then, the solving equations four these four straight cable segments $B_{xi}$-$B_{x1}$, $B_{x1}$-$B_{x2}$, $B_{x2}$-$B_{x3}$, and $B_{x3}$-$B_{xi+1}$ can be respectively established in a similarly way, and their discrete models and balanced forces are shown in Figure 6. Taking the first straight cable segment $B_{xi}$-$B_{x1}$ as an example, by considering the equilibrium conditions the governing equations are established as:

$$L_{x1} = \frac{H_{gj-1}}{T_{gj-1}} S_{x1}, \tag{9}$$

$$h_{x1} = \frac{V_{gj-1}}{T_{gj-1}} S_{x1}, \tag{10}$$

$$T_{x1} = \sqrt{V_{x1}^2 + H_{x1}^2}, \tag{11}$$

$$H_{gj-1} = H_{x1}, \tag{12}$$

$$V_{x1} = V_{gj-1} - q_2 S_{x1} - W_1, \tag{13}$$

$$Y_{x1} = h_{x1} + Y_{gj-1}, \tag{14}$$

where $L_{x1}$ and $h_{x1}$ are horizontal length and vertical heights between the two nodes $B_{xi}$ and $B_{x1}$; $S_{x1}$ is the arc length of $B_{xi}$-$B_{x1}$; $V_{x1}$; $H_{x1}$ are the horizontal and vertical forces at node $B_{x1}$; $T_{x1}$ is the tension vertical force of the rail-cable segment $B_{xi}$-$B_{x1}$; $Y_{x1}$ is vertical coordinate of the node $B_{x1}$.

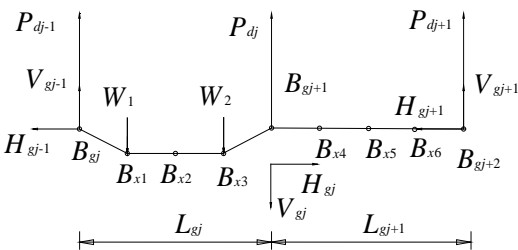

**Figure 5.** Analytical chart of rail-cable segment that the shifting-girder trolley is moving on.

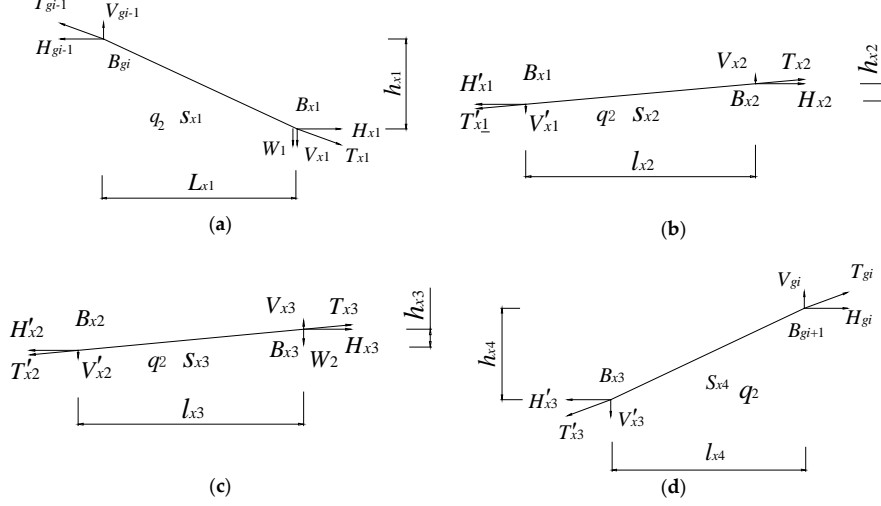

**Figure 6.** Discrete model of the four straight cable segments for the rail-cable segment where the shifting-girder trolley is moving on: (**a**) segments $B_{xi-1}$-$B_{x1}$; (**b**) segments $B_{xi}$-$B_{x1}$; (**c**) segments $B_{x1}$-$B_{x2}$; (**d**) segments $B_{x2}$-$B_{xi}$.

*2.3. Solution to the Shifting-Girder System*

In the shifting-girder process, the material, diameter and pre-tension of the rail-cable are known. The rail-cable anchor position $L_g$, pre-tension force $T_{g0}$, stress-free length of the sling $S_{di}$, main-cable force, cable shape, distance $L_z$ and height difference $h_z$ of the towers can be obtained by resetting the calculation from the bridge's service state to the specific erection state of the girder. Next, other unknown variables can be solved by the iterative process shown in Figure 7, where $\Delta_{l1}$, $\Delta_{h1}$, and $\Delta_{g1}$ are convergence conditions and $\varepsilon$ is the given error limit. According to this iterative procedure, the relative deformation of the rail-cable and the relative coordinates of each lifting points under shifting-girder process can be obtained, then the systematical response can be finally determined. To more accurately investigate the variation of the local deflection curve of rail-cable, the rail-cable segment where the shifting-girder trolley is moving on is simulated by more detailed straight-rod elements. Also, the variation of system response is added to the local area to analyze the local deflection of the rail-cable.

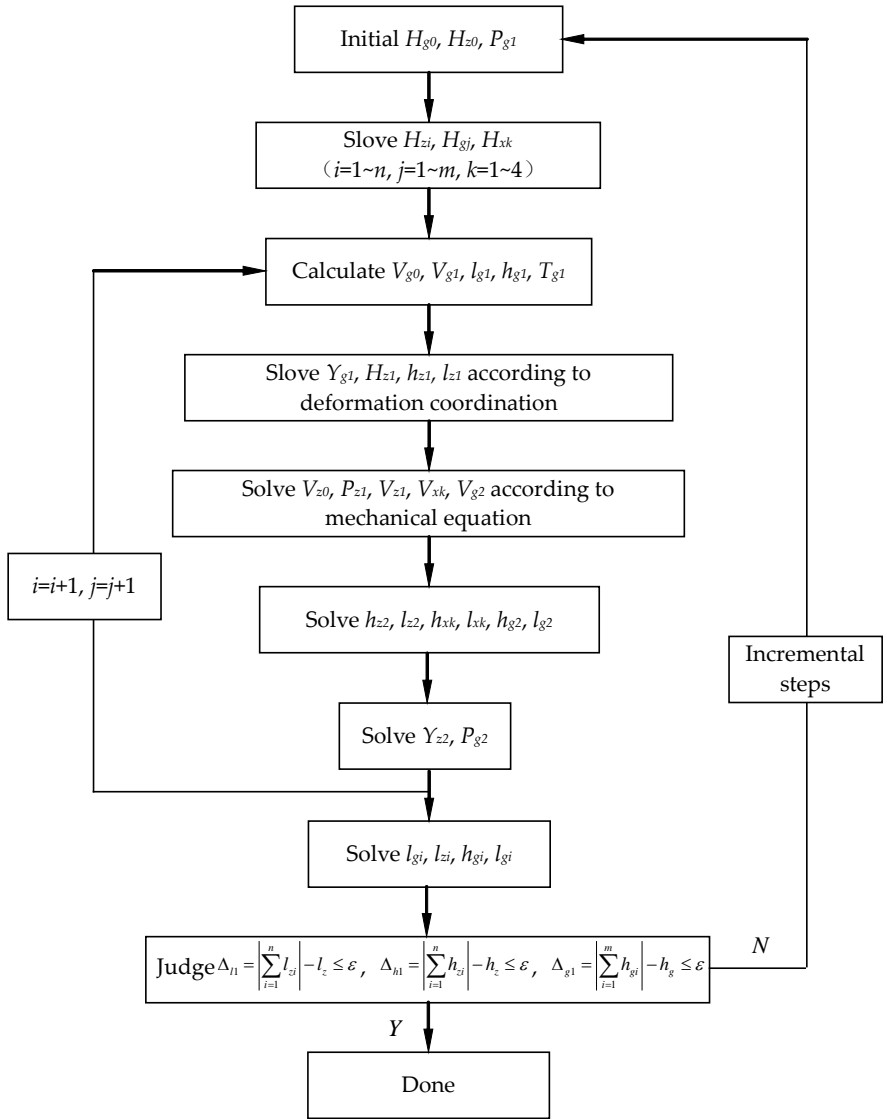

**Figure 7.** Iterative flow chart of shifting-girder process undergoing girder loads.

## 3. Global Model Test of Shifting-Girder System Undergoing Girder Loads

The girder erection is the most important stage of the superstructure construction for a suspension bridge. The RCSG technique is a newly developed technology for the construction of main girder

of a suspension bridge, which directly affects the construction progress and quality. Therefore, to verify the correctness of the presented theoretical method, a global indoor test model with a 1:33 reduced-scale geometric ratio is built in the Structure Center of Changsha University of Science and Technology, by taking the girders erection of the Aizhai suspension bridge as the engineering background. The model span of the main-cable is 3.515 + 35.636 + 7.333 + 1.942 m, as shown in Figure 8. The test model [19] is composed of an anchorage system, main-cable, tower, rock anchor cable [22], sling and cable clamp, shifting-girder system and measuring equipment. In the laboratory, the tower and main-cable anchorage model were installed, the empty cables were erected and adjusted, and the cable clamps, slings, hanging saddles, and tensed rail-cables were installed. The counterweight was used to simulate the movement and placement of the cross-cable crane, the dead weight of shifting-girder trolley, and the transportation and placement of girder segments.

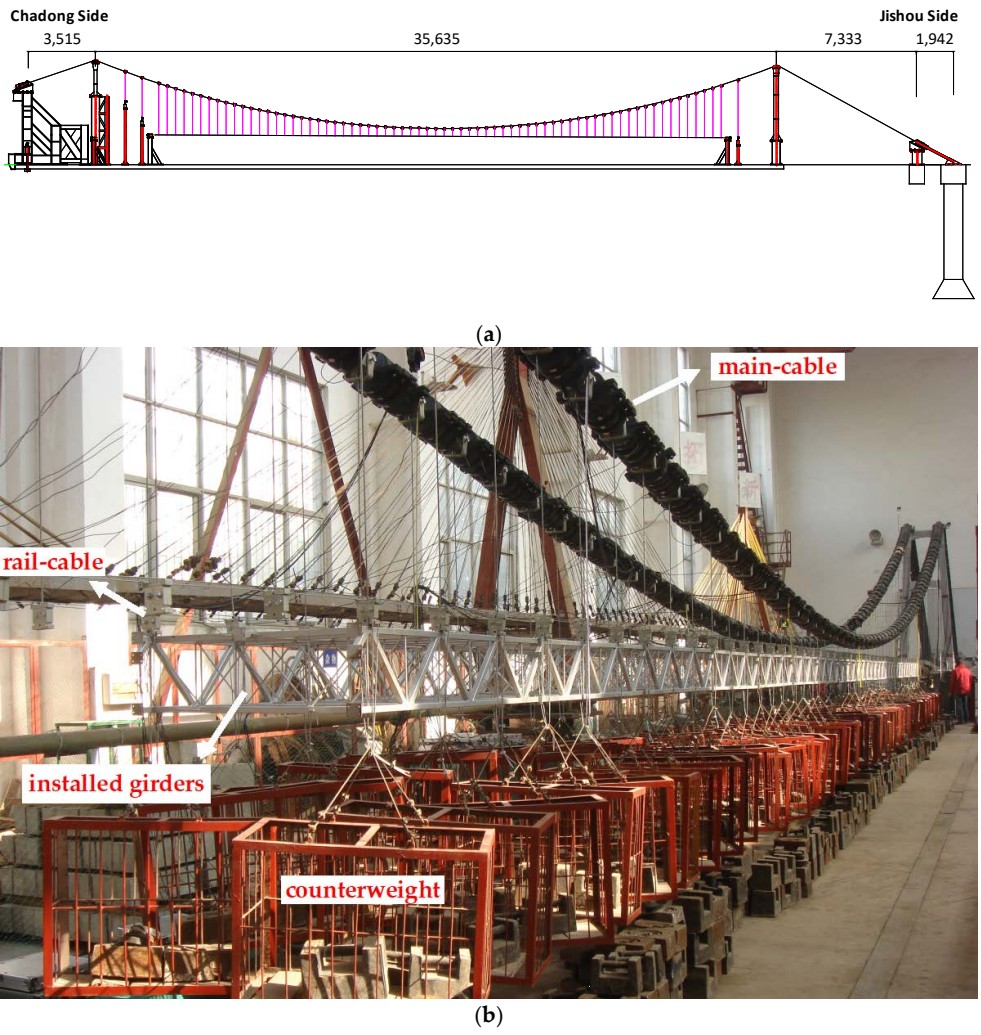

**Figure 8.** The global indoor test model of the Aizhai suspension bridge: (**a**) elevation view (unit: mm); (**b**) on-site testing.

There are totally 69 steel truss girder segments labeled as: No. B36 is the mid-span girder segment in the centre span; Nos. B35–B1 are the standard girder segments at the Jishou side; Nos. B33′–B1′ are the standard girder segments at the Chadong side. During construction, the mid-span girder segment B36 was first moved and installed; then B35 and B33′ girder segments are respectively connected with B36 at the Chadong side and the Jishou side; next, B34 and B32′ are respectively connect with B35 and B33′, and so on, until the completion of erecting the global bridge. Twenty-one working

cases were designed from the erection of empty cable to accomplishment of the girder segments in the global model test, and specific working cases are shown in Table 1. To more precisely test the experimental results, several test systems are fixed up and multiple intelligent sensors [32–34] are embedded. The tower deviation, main-cable deformation, main-cable force, rail-cable force, rail-cable shape, and sling force are tested under all the design cases. The test results are compared with the theoretical results, to validate the presented method can correctly solve the pseudo-static analysis of the RCSG system or not.

**Table 1.** Working cases of global model test of the Aizhai suspension bridge.

| Working Cases | Contents | Working Cases | Contents |
| --- | --- | --- | --- |
| 1 | adjusting empty cables | 12 | erection of girders B19–21, B17'–19' |
| 2 | tension of rock-anchor cable | 13 | erection of girders B18, B16' |
| 3 | tension of rail-cable | 14 | erection of girders B17, B15' |
| 4 | erection of girders B36 | 15 | erection of girders B16, B14' |
| 5 | erection of girders B35, B33' | 16 | erection of girders B13–15, B11'–13' |
| 6 | erection of girders B34, B32' | 17 | erection of girders B10–12, B8'-10' |
| 7 | erection of girders B31-33, B29'–31' | 18 | erection of girders B7–9, B5'–7' |
| 8 | erection of girders B28-30, B26'–28' | 19 | erection of girders B4–6, B2'–4' |
| 9 | erection of girders B25-27, B23'–25' | 20 | erection of girders B3, B1' |
| 10 | erection of girders B24, B22' | 21 | erection of girders B2, B1 |
| 11 | erection of girders B22–23, B20'–21' | | |

## 4. Discussion on Results

The pseudo-static results are shown in Figures 9–12 for the 21 working cases listed in Table 1, by comparing the presented theoretical method and the measured experimental value of the model test. From these figures, it can be learned that that the forces of the main-cable, displacements of the main-cable, horizontal deflections of the two towers and forces of the rail-cable in all working conditions by the presented theoretical method and the indoor-model test method all have the same distribution regularities, and the difference of all the mentioned results for these two methods is small, mostly less than 5%. Therefore, the theoretical results agree well with the results from the model test.

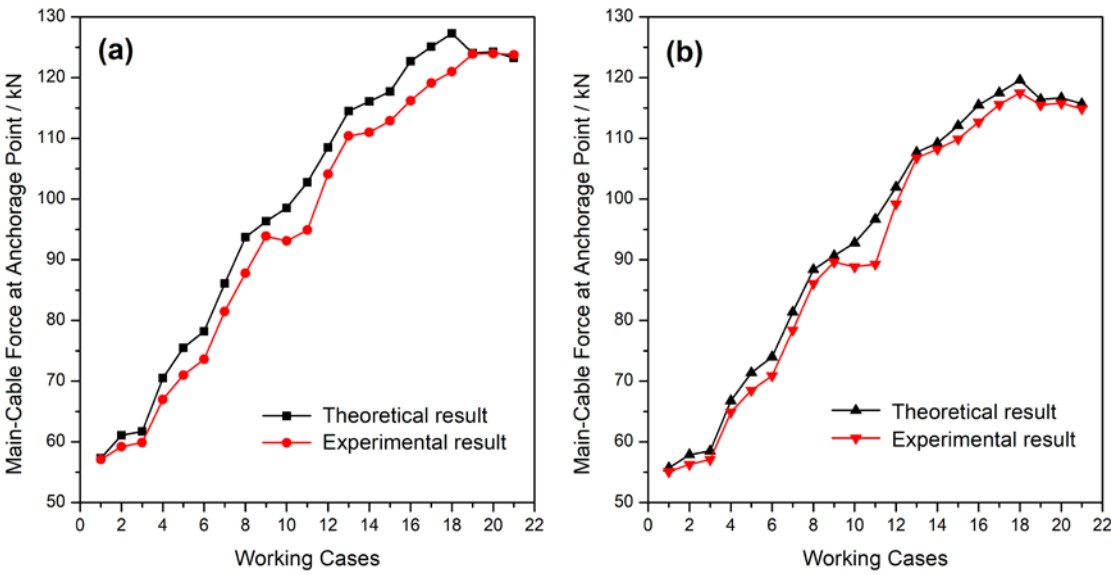

**Figure 9.** Comparison of the main-cable forces at the anchorage point by the presented theoretical result and experimental result on different shifting-girder working cases: (**a**) Jishou side; (**b**) Chadong side.

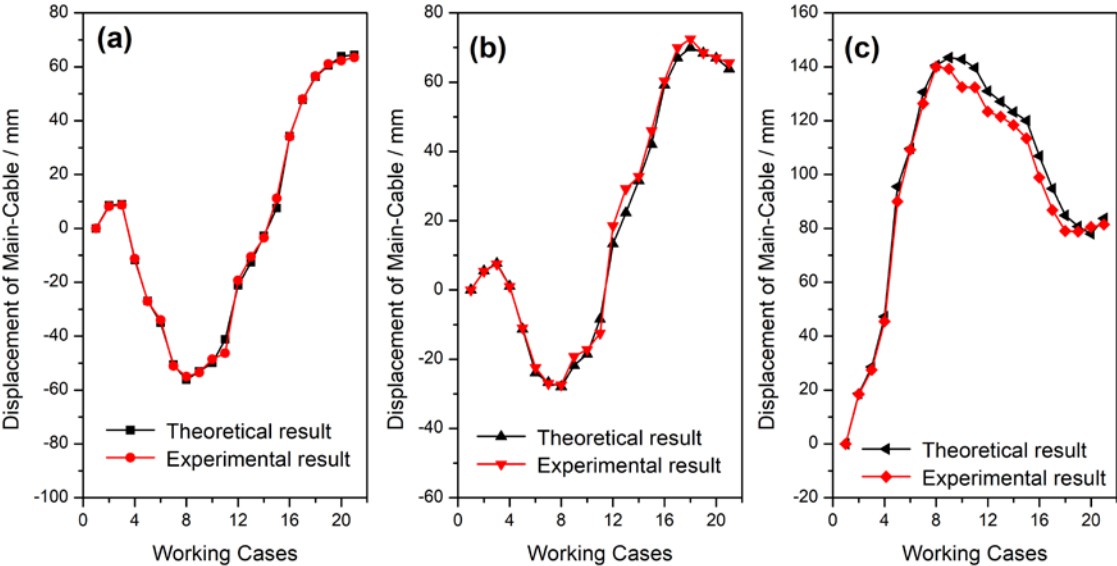

**Figure 10.** Comparison of displacement for main-cable by the presented theoretical result and experimental result on different shifting-girder working cases: (**a**) J14 cable at Jishou side; (**b**) C15 cable at Chadong side; and (**c**) mid-point of main span.

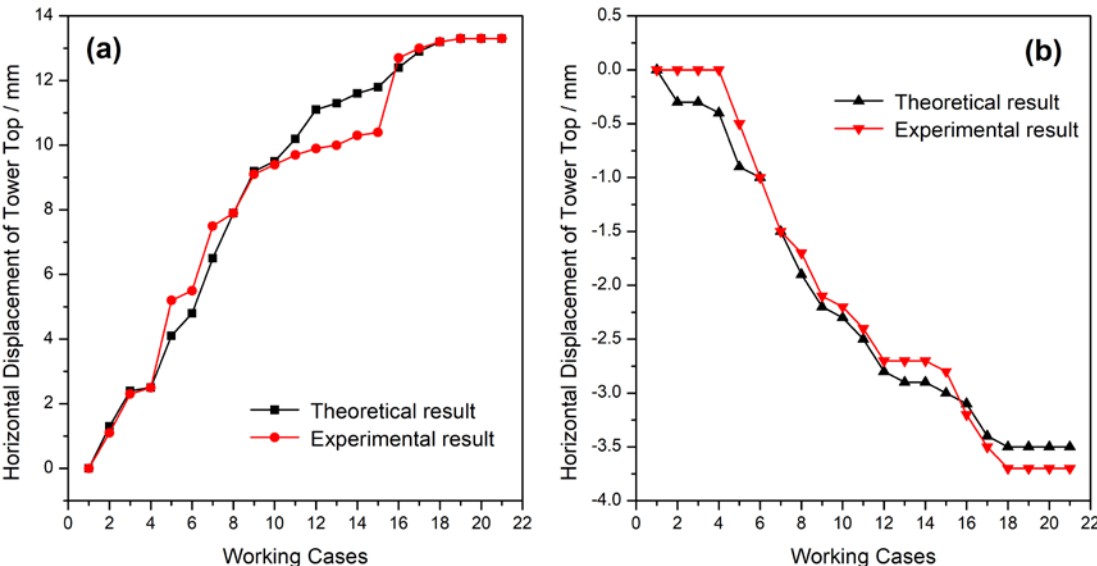

**Figure 11.** Comparison of the horizontal displacement of tower top along span by presented theoretical result and experimental result on different shifting-girder working cases: (**a**) Jishou side; (**b**) Chadong side.

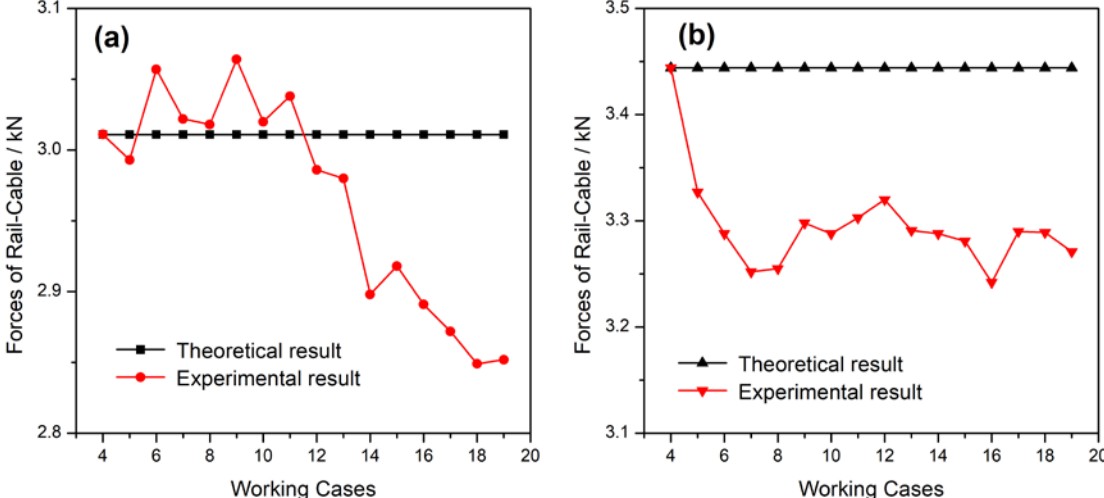

**Figure 12.** Comparison of rail-cable forces by the presented theoretical result and experimental result on different shifting-girder working cases: (**a**) Jishou side; (**b**) Chadong side.

Figure 9 shows the forces of the main-cable at the anchorage point under the 21 working cases. It can be learned that the forces of the main-cable at the anchorage points at both Jishou and Chadong sides are increasing, with the erection of the girder segment. The variation of the incremental cable forces is small when the subsequent girder segments are erected, only a slight decline is observed at the installation stage of the end girder segment. This is related to the temporary loads which includes the removing of the rail-cable and unloading of the shifting-girder trolley.

Figure 10 shows the deflections of the main-cable on different working cases. From the beginning working case, the variation of the deflection for the main-cable at sling J14 (i.e., l/4 main span $L_z$) first decreases and then increases, and reaches the maximum value when the erections of the girder segments are completed. Similarly, the variation of the deflection for the main-cable at sling C15 (i.e., 3/4 main span $L_z$) first decreases and then increases. Obviously, this similarity between the two main-cable's deflections at sling J14 and sling C15 is due to the symmetry of the global structure along the mid-span in the direction of main span. Affected by the two-pair rock anchor cable [22] at the Chadong side and the installation position of the girder segments, the deflection of the main-cable at sling C15 reaches its maximum value when the B5' girder segment is being installed. Moreover, as shown in Figure 10c, the variation of the deflection for the main-cable at the mid-point of the main span increases first and then decreases, with the progress of erecting of the girder segments. The deflection for the main-cable at the mid-span reaches the maximum when the erections of B25–27 and B23'–25' are completed, i.e., 1/4 of the girders of the global suspension bridge is completed. Next, this deflection gradually decreases, while the variation rate is small. When the number of completed girder segments is bigger, the gravity stiffness of the main-cable is greater. And the influence of the installation of the subsequent girder segments on the main-cable's deflection at mid-point becomes smaller and smaller.

Figure 11 illustrates the comparison of the horizontal displacement of the tower between the presented theoretical results and measured values at both sides for the 21 working cases. With the erection of girder segments, the tower deflections at both the Chadong and Jishou sides are increasing. And the two horizontal deflections are both pointing to the mid-span at all the erecting time. Moreover, the variations of the two towers' horizontal deflections are small at the erecting stages of the last several girder segments.

Figure 12 shows the comparison of the rail-cable forces between the presented theoretical results and measured values for the 21 working cases. With the erection of the girder segments, the measured rail-cable force has a slight deviation from the theoretical value in each working case. While the variation range is not large, the deviation is basically less than 5%, and, in a few working cases,

it reaches the maximum deviation of 6%. This difference is acceptable from engineering viewpoint, which indicates these two methods match well with each other.

Besides the forces of the main-cable, displacements of the main-cable, horizontal deflections of the two towers, and forces of the rail-cable mentioned in Figures 9–12, the theoretical results of the shape of the rail-cable, sling force, etc. are all in good agreement with experimental results for the 21 working cases. The compared results show the calculated values by the presented method for the deflection and force of main-cable, sling force, deformation of rail-cable match well with the tested values during the construction of girder segments. Therefore, the presented theoretical method can correctly and effectively obtain the forces and deflections of the RCSG system undergoing girder loads, and the systematical responses are conform to the prediction values, thus the correctness of the presented method is validated. Furthermore, in the shifting-girder process undergoing girder loads, the theoretical results and the measured results for the vertical deflection of rail-cable and the force at anchor points are basically consistent. This indicates that the presented method truly reflects the actual mechanical performance of the structure, and the deformation and force of rail-cable can be studied and solved by the proposed theory in this research. Additionally, with the erection of the girder segments, the measured force of the rail-cable has a slight deviation from the theoretical value in each working case. Nevertheless, the deviation range is not large and the deviation value basically stays within 5% with a maximum deviation of 6% in only a few of the working cases. When the gravity center of the girder segment reaches the mid-point of inter-node, the relative deformation of the rail-cable is the largest. While the increase of the erected girder segments will not directly affect the local deformations of the rail-cable. Generally speaking, the correctness of the presented method is validated by the experimental results, and it is able to effectively obtain the reflections of the shifting-girder system undergoing girder loads when the long span suspension bridge adopts the RCSG technique for construction.

## 5. Conclusions

This paper is focused on the pseudo-static responses of the shifting-girder system of the new erecting method of the main girders (i.e., RCSG technique) for the long span suspension bridge in a rural mountain area with poor transportation and no navigable rivers available for carrying large components. The global mechanical model of the double-layer cable system in the shifting-girder process is established. The solving program is developed to obtain the forces and deflections of the main cable, rail-cable, and slings when the shifting-girder system is undergoing the girder segments' weight. By taking the Aizhai suspension bridge as the engineering background, a global indoor reduced-scale model of a shifting-girder system is also designed to validate the presented theoretical method. The results for the presented method and model test are compared. The conclusions are drawn as follows:

(1)  The theoretical results match well with the measured experimental results of the global model test for the 21 working conditions of the shifting-girder system undergoing girder loads. The differences of the forces and deflections from these two methods are small for most of the working conditions, except the maximum deviation of the measured force of rail-cable from the theoretical value reaches 6% in only a few working cases.

(2)  The main-cable's forces at the anchorage points on both sides are increasing in the erecting process of girder segments. The variation of the incremental cable forces is small in the erecting courses of the subsequent girder segments.

(3)  The vertical deflections of the main-cable at the locations of both l/4 and 3/4 main span first decrease and then increase in the erecting process, and reach the maximum value when the erections of the girder segments are completed, due to the symmetry of the global structure along the mid-span. The deflection of the main-cable at mid-point first increases and then decreases in the erecting progress, and it reaches the maximum value when 1/4 of girders are completed.

(4) The horizontal deflections of the two towers are increasing at all erecting times, and both point to the mid-span. The variations of these two deflections are small at the erecting stages of the last several girder segments.

(5) The measured rail-cable force of the model test has a slight deviation from the theoretical value, while the variation is basically less than 5% and reaches a maximum of 6% in a few working cases.

(6) The correctness and effectiveness of the presented theoretical method is validated by the experimental results of indoor model test. It is able to obtain the forces and the reflections of the double-layer cable system of the shifting-girder system undergoing girder loads. The accuracy and results satisfy the requirement of practical engineering application, and this simplified method is suitable to solve the RCSG system under load-condition for the long span suspension bridge adopting the RCSG technique for construction. Also, the modeling and studying of the presented theoretical method can supply references for the similar double-layer projects carrying loads.

**Author Contributions:** Conceptualization, Q.P. and D.Y.; Methodology, Z.Y.; Software, Q.P. and Z.Y.; Validation, H.X.; Formal Analysis, Q.P.; Investigation, Q.P. and D.Y.; Resources, D.Y.; Data Curation, Q.P. and D.Y.; Writing—Original Draft Preparation, Q.P. and Z.Y.; Writing—Review & Editing, Q.P. and Z.Y.; Visualization, Z.Y.; Supervision, D.Y.

**Funding:** This research was funded by the National Natural Science Foundation of China (Grant No.51678069); the Natural Science Fund of Hunan Province (Grant No. 2018JJ2436); the Key Discipline Fund Project of Civil Engineering of Changsha University of Sciences and Technology (Grant No. 18ZDXK02, 18ZDXK14); Hunan Provincial Education Department Innovation Platform Fund (Grant No. 18K046); and the innovation team of the safety assessment theory and performance technology of the long span bridge of Changsha University of Science and Technology.

**Acknowledgments:** The authors thanks to Gangbing Song of University of Houston for his very helpful remarks and comments.

**Conflicts of Interest:** The authors declared that they have no conflict of interest to this work.

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
