# Peer review of "Pseudo-Static Analysis on the Shifting-Girder Process of the Novel Rail-Cable-Shifting-Girder Technique for the Long Span Suspension Bridge"

_applsci, doi:10.3390/app9235158_

Round 1

Reviewer 1 Report

The paper presents the results of the substantial research on the important and relevant problem. The conducted investigations are relevant and original. The description of the mechanical model for the shifting-girder system undergoing girders loads is detailed and clear.

English needs minor improvement, e.g.:

row 150 - (2) The stress-strain … to the Hooke's law

row 190 – segment catcatric element

rows 66, 114, 115, 163, etc. – it is not clear, why the dash is needed. E.g., in the sentence “… model of the rail-cable shifting-girder system…” (row 160) the “rail-cable” is the adjective and using dash is logical, however in “… hanging saddles and tensed-rail-cables were installed” (row 246) it is the subject.

Few remarks (probably, considering future studies):

(1) There would be useful to perform the economical comparison of the RST with other shifting techniques for the long span suspension bridges.

(2) It might be useful to provide engineers with the simplified calculation method, the strict design technique elaborated on the basis of the presented global mechanical model.

(3) Considering the assumptions of the mechanical model, it would be interesting to work on the technique considering the non-Hooke's law cables (providing plastic behavior or yielding).

Author Response

Comments and Suggestions for Authors

The paper presents the results of the substantial research on the important and relevant problem. The conducted investigations are relevant and original. The description of the mechanical model for the shifting-girder system undergoing girders loads is detailed and clear.

Answer: The authors thank Reviewer 1’s comments and suggestions.

Q1. English needs minor improvement, e.g.:row 150 - (2) The stress-strain … to the Hooke's law

A1. “The Hooker theorem” here is replaced by the right one "the Hooke's law”.

Q2.row 190 – segment catcatric element

A2. “segmented catcatric element” here has been rewritten as " the segment catenary element”.

Q3.rows 66, 114, 115, 163, etc. – it is not clear, why the dash is needed. E.g., in the sentence “… model of the rail-cable shifting-girder system…” (row 160) the “rail-cable” is the adjective and using dash is logical, however in “… hanging saddles and tensed-rail-cables were installed” (row 246) it is the subject.

A3. The dash “-” is used to explain the nouns of rail-cable and this new girder erecting technique. By adopting the suggestion from Reviewer 1, we have unified the expressions as rail-cable-shifting-girder (RCSG), main-cable, rail cable, etc. And “tensed-rail-cables” in row 246 has revised as “tensed rail-cables”, here rail-cable is a noun being subject.

Few remarks (probably, considering future studies):

(1) There would be useful to perform the economical comparison of the RST with other shifting techniques for the long span suspension bridges.

(2) It might be useful to provide engineers with the simplified calculation method, the strict design technique elaborated on the basis of the presented global mechanical model.

(3) Considering the assumptions of the mechanical model, it would be interesting to work on the technique considering the non-Hooke's law cables (providing plastic behavior or yielding).

Answer: These remarks and suggestions are very helpful for our future research. Thanks!

Reviewer 2 Report

The reviewer finds this manuscript very interesting to practical engineering applications. Also, the small lack of scientific approach can also be noticed.

There are some grammatical and linguistical errors that need to be rectified. For example, there are some oddly shaped phrases in lines 77, 78 and 94. 

I'm not sure what is the meaning of "permanent steel" in line 109 and "safeties" in subsequent line. Can authors comment this? Please check also lines 135, 139 and 293. There are probably more lines like this, so it's advisable to check them thoroughly.

Furthermore, in line 150...did authors mean the Hooke's law? Also, the 5th point of subsection 2.1, states that the friction between the rail-cable and hanging saddle is ignored. Can authors give some additional explanation to why the friction can be ignored? 

Regards!

Author Response

Comments and Suggestions for Authors

The reviewer finds this manuscript very interesting to practical engineering applications. Also, the small lack of scientific approach can also be noticed.

Answer: The authors thank Reviewer 2’s comments and suggestions.

Q1. There are some grammatical and linguistical errors that need to be rectified. For example, there are some oddly shaped phrases in lines 77, 78 and 94.

A1. These some grammatical and linguistical errors in lines 77, 78 and 94 have been revised try our best.

Q2. I'm not sure what is the meaning of "permanent steel" in line 109 and "safeties" in subsequent line. Can authors comment this? Please check also lines 135, 139 and 293. There are probably more lines like this, so it's advisable to check them thoroughly.

A2. The “permanent steel" in line 109 is the steel using of the girder, and this Chinese expression has been rewritten as English style “steel”. The "safeties" has not mentioned in this paper, so it is deleted here. Including lines 135, 139 and 293, the whole paper has been check carefully and the revised parts are denoted by red color.

Q3. Furthermore, in line 150...did authors mean the Hooke's law? Also, the 5th point of subsection 2.1, states that the friction between the rail-cable and hanging saddle is ignored. Can authors give some additional explanation to why the friction can be ignored?

A3. Yes, here is the Hooke's law and it is replaced by the right one. The additional explanation on why “the friction between the rail-cable and hanging saddle is ignored” is added in the 5th point of subsection 2.1. It is due to the fact that the hanging saddle is smooth and their relative deformations are small.

Round 2

Reviewer 1 Report

Dear Authors,

Thank you for the revised version.

I have no more questions.

Best regards.

Author Response

Thank Reviewer 1’s comments and suggestions.

Reviewer 2 Report

Please, once more check the complete paper for grammatical errors. Also, I think that acknowledgment to the reviewers at the end of the paper is not necessary.

Regards!

Author Response

Thank Reviewer 2’s comments and suggestions. We have tried our best to check the whole paper for grammatical errors thoroughly. Also, the acknowledgment to the reviewers is deleted.